# Detecting Hallucinations in LLM Responses Using Token-Level Log-Probability Signals

**Vadim Eliseev & Aleksandra Maksimova**
Laboratory of Intellegent Systems
Institute of Applied Mathematics and Mechanics
Donetsk, Russian Federation
{eliseevv02,maximova.alexandra}@mail.ru

## Abstract

Large language models (LLMs) have proven themselves to be powerful tools for many natural language tasks — from being a high-quality text classifiers to acting as agents in complex retrieval-augmented generation (RAG) systems. However, from early beggining they suffer from a major limitation: hallucinations, i.e. confidently generating incorrect or misleading information that can also slightly correlate with the given task. This issue is critical in error-sensitive domains such as finance, medicine, and law, where even small inaccuracies can cause significant harm and detriment. In this study we address the early detection of hallucinating answers based on user input (prompt), answer by the LLM, and which is more important — token-level probabilty signals that can also be extracted from the LLM during its inference time. We constructed a dataset that combines textual information with sequences of token log-probabilities and their statistics (mean, min, variance, percentiles, etc.), labeled the answers whether they are hallucinations or not. We trained a lightweight classifier that outputs the probability that a given response is a hallucination. We evaluate the classifier and perform ablation studies to quantify the contribution of token-level signals versus text-only features. The intended use of the trained model is to be a standalone output guard agent in multi-agent system that rejects the answer of LLM-generator if its hallucination probability is above acceptance threshold and protects the users of it from having incorrect or misleading answer by making the whole system regenerate such answer or confirm that it cannot give the faithfull reply.

## 1 Introduction

Continuously advancing large language models (LLMs) such as LLaMA Touvron et al. (2023) family, newest versions of GPT Singh et al. (2025), Gemini Team et al. (2023) and others have demonstrated strong performance in a wide range of tasks beyond theoretical science and laboratory conditions. They are increasingly deployed in real-world applications, including customer support automation Landolsi et al. (2025), financial document analysis, legal assistance Xi et al. (2025), medical information systems Hassan et al. (2025), and retrieval-augmented enterprise workflows in both single and multi-agent approach Li et al. (2024).

Modern multi-agent systems are capable not only of answering textual queries using corporate knowledge data via classical retrieval-augmented generation (RAG) Lewis et al. (2020), but also of interacting with external tools and services through function calling and structured output generation, powered by ReAct Yao et al. (2022) approach. Instead of producing free-form text responses alone, contemporary LLM-based agents can generate machine-readable outputs (e.g., JSON schemas, API calls, database queries) that trigger downstream actions within larger software ecosystems. This shift transforms LLMs from purely generative text models into decision-making components embedded within executable workflows, enabling automation of complex business processes.

However, one of the most significant limitations of LLMs – **hallucinations** – persists even in state-of-the-art models.They hallucinating because their training and evaluation procedures reward guessing over acknowledging uncertainty Kalai et al. (2025). This issue becomes critical in real-world

applications, where LLMs playing role of a decision makers, especially when it comes about high-stakes domains such as finance, law or medicine. In this case, timely detection and mitigation of hallucinated responses can substantially reduce risks for both businesses and end users.

In this study we address the early detection of hallucinating LLM responses, focusing on applying proposed solution as an output guardrail agent within a multi-agent system. As the latency is a crucial constraint in real-world multi-agent deployments, we focus on building a lightweight hallucinations detection model that can be integrated into such systems without introducing substantial computational or latency overhead.

During the practical part of our study the following steps were performed:

1. Adopted an existing dataset containing user prompts and model-generated responses with additional features, initially lacked annotations for hallucinations;

2. Annotated the dataset with a binary label (0 or 1) indicating whether a given model response constitutes a hallucination;

3. Analyzed the labeled dataset, extracted the suitable for classification features from token log-probabilities;

4. Trained the basic versions of classical ML algorithms on labeled dataset in order to find the best baseline classifier;

5. Performed hyperparameter tuning of the best classifier from previous step in order to improve model performance.

These steps will be described in the following sections, the source code of the project is open-source and available at:

```
https://github.com/EliseevVadim/advanced-hallucinations-detection
```

## 2 Related Work

Hallucinations in LLMs have been a subject of extensive research from both theoretical and applied perspectives. Early work highlighted that LLMs may produce fluent yet factually incorrect or unsupported content, often with high confidence Ji et al. (2023). Subsequent studies analyzed the underlying causes of hallucinations, including exposure bias, limitations of next-token prediction objectives, and misalignment between training objectives and factual correctness Kalai et al. (2025). The hallucinations taxonomy provided later at Huang et al. (2025) shows the different types of hallucinations combined with their key reasons that can help engineers mitigate them more precise.

In practical side, hallucination mitigation has been approached from several complementary directions. One of the most common strategies is retrieval-augmented generation (RAG), which grounds model outputs in external knowledge sources to improve factual consistency Lewis et al. (2020). However, RAG systems are still hallucinating, so this cannot be used as a "silver bullet" as it is. Another line of work focuses on post-generation verification, where either rule-based systems or secondary LLMs act as judges to assess factual correctness and consistency Kadavath et al. (2022). LLM-based hallucination detection pipelines are also discussed at Anaokar et al. (2025). Other frameworks such as HuDEx integrate hallucination detection with explainability, enabling models not only to flag problematic outputs but also provide interpretable error explanations Lee et al. (2025). However these approaches might not be perfect in real-world scenarios as it invokes another LLM that should be at least as good or better than our generator which causing computational and latency overhead. Also LLM-as-a-judge pipelines considering only textual inputs, but in practice we can obtain also some numerical internal information about model reply, such as token probabilities (which we are focusing in current study), internal representations from LLMs layers, etc.

Lightweight neural models like HaluNet Tong et al. (2025) leverage multi-granular uncertainty modeling, fusing token probabilities with semantic embedding information to deliver efficient one-pass detection suitable for real-time applications. Other approaches address both detection and calibration by capturing hallucination signals within output and semantic spaces, such as HaDeMiF Zhou et al. (2025), which uses compact decision trees and neural networks to calibrate and mitigate hallucinations in models outputs during inference. While these approaches have demonstrated

improvements in factual robustness, they still introduce additional computational overhead, require external knowledge sources, or depend on auxiliary models, which may limit their suitability for latency-sensitive multi-agent systems.

# 3 METHOD

To address the incompleteness of LLM-based hallucinations detectors (as they process textual inputs only) and complexity of systems, based on neural networks, combined with LLMs we propose a lightweight hallucination detection framework based on classical machine learning algorithms. This enables us to take into account both textual features and internal generation information (e.g., token-level probabilities and model hidden states).

In order to do this we utilized the publicly available dataset Massaron (2024) which contains responses generated by Mistral 7B Instruct Jiang et al. (2023) to prompts from the Open Orca dataset Lian et al. (2023). In addition to textual information, the dataset includes structured generation metadata such as the number of tokens in the response (according to Mistral 7B tokenizer), token-level probabilities, and the generation finish reason. The dataset in initial form does not contain hallucination labels. Therefore, an explicit annotation procedure was required prior to supervised training. The preprocessing and labeling pipeline is described in detail in Section 4.

Following dataset annotation, we constructed a collection of baseline classifiers using classical machine learning algorithms initialized with default hyperparameters. This stage was designed to identify the most suitable model for the task independently of hyperparameter optimization. For textual representation, we employed sparse vectorization techniques which performing effective in combination with linear classifiers for text classification tasks Joachims (1998). Sparse representations are often preferable for classical ML pipelines, as dense neural embeddings may not provide consistent advantages outside deep architectures and can be suboptimal for linear models Reimers & Gurevych (2019). We evaluated different combinations of textual encoders and classification algorithms to determine the most effective detection pipeline. The procedure is described in Section 5.1.

After selecting the best classification pipeline the hyperparameter optimization was applied. We identified the most influential parameters of the chosen classifier and defined appropriate search ranges. A search procedure was then applied to determine the optimal hyperparameter configuration according to the selected evaluation metric. Details of the hyperparameter tuning stage and final configuration are provided in Section 5.2.

# 4 DATA PREPARATION AND ANALYSIS

This section outlines the preprocessing, exploratory data analysis, and labeling procedures applied to the dataset making it suitabile for supervised training of classification pipelines.

## 4.1 RAW DATA PREPROCESSING

As the source of training data we utilized the publicly available dataset Massaron (2024). It is based on Open Orca dataset Lian et al. (2023) and contains responses generated by Mistral 7B Instruct Jiang et al. (2023) to its prompts. The dataset contains following attributes: the original prompt, the reference answer, the model-generated answer, decoding parameters (temperature and top-$p$), the number of completion tokens (i.e., the length of the generated response in tokens), token log-probabilities of each generated token, and the generation finish reason (indicating whether the generation terminated naturally or was truncated due to the maximum output length constraint).

After an initial exploratory analysis of the dataset, we observed that each prompt is associated with **21** generations, resulting in a total of **392,091** records. Such a large number of responses per prompt is redundant and will cause computational overhead during the labeling process, so we decided to reduce its size. To achieve this while preserving informative variability, we introduced a scalar *risk score*– a proxy feature designed to estimate the likelihood that a given generation may contain hallucinated content. This feature was used only for dataset size reduction and was removed in all subsequent training stages.

The *risk score* was computed using the token log-probabilities of each generated response. Let $\{\ell_i\}_{i=1}^n$ denote the log-probabilities of the generated tokens, where $n$ is the number of tokens in the response.

We define

$$\mu = \frac{1}{n}\sum_{i=1}^{n}\ell_i \tag{1}$$

as the mean log-probability,

$$\sigma = \sqrt{\frac{1}{n}\sum_{i=1}^{n}(\ell_i - \mu)^2} \tag{2}$$

as the standard deviation of log-probabilities,

$$f = \frac{1}{n}\sum_{i=1}^{n}\mathbf{1}(\ell_i < \tau) \tag{3}$$

as the fraction of low-confidence tokens, where $\tau$ denotes a predefined confidence threshold and $\mathbf{1}(\cdot)$ is the indicator function.

The risk function is then defined as

$$\mathrm{Risk}(\{\ell_i\}) = \frac{-\alpha\mu + \beta\,\sigma + \gamma\,f}{\log(n+1)} \tag{4}$$

where $\alpha, \beta, \gamma \geq 0$ are tunable coefficients that control the contribution of each component to the overall risk score.

In the current implementation, the coefficients were set to $\alpha = 1$, $\beta = 0.5$, $\gamma = 0.3$ and $\tau = -5.0$.

The components of the risk function can be interpreted as follows:

- $-\mu$ increases the risk when the model's average confidence is low (the negative sign ensures that lower mean log-probability results in higher risk);
- $\sigma$ increases the risk when the token-level predictions exhibit high variability;
- $f$ amplifies the contribution of tokens with particularly low confidence;
- division by $\log(n+1)$ normalizes the risk with respect to response length.

Additionally, we observed that all generations were produced with a fixed top-$p$ parameter of 0.95, while the temperature took only two values: **0.0** or **0.8**. Based on this observation, we computed the risk score for each generation and performed group-wise selection within each prompt according to the following criteria:

- Row with temperature = 0.0 as it is most deterministic one;
- Row with lowest risk score in the group (most likely not a hallucination);
- Row with biggest risk score in the group (most likely a hallucination);
- Row with closest to median risk score in the group (most interesting examples for classification).

Applying this risk-based filtering strategy reduced the dataset size to **69,451** records, which is approximately six times smaller than the original dataset.

Finally, we removed records in which the generated answer did not contain any numeric characters or alphabetic symbols (Latin or Cyrillic), as such outputs were considered non-informative for the classification task. This additional filtering step eliminated another **1,056** records.

## 4.2 DATA LABELING

After reducing the dataset size, the remaining samples were labeled whether each generated response contains hallucinated content. We adopted a semi-automatic labeling strategy based on two complementary quantitative metrics.

For each record, we computed the following scores:

- $h \in [0,1]$ – the `hallucination_score`, obtained from the Google Gemma-3 12B model Kamath et al. as a judge via using the prompt shown in Figure 4.2;

- $s \in [0,1]$ – the `similarity_score`, representing the cosine similarity Salton et al. (1975) between the generated answer and the reference answer. Vector representations of both answers were obtained using *nomic-embed-text-v1.5* model Nussbaum et al. (2024)

---

*SYSTEM*
You are a world-class state-of-the-art assistant for determining whether a Model Answer is a hallucination / contains hallucinations or not based on a Prompt or an Expected Answer.
Your task is to estimate the likelihood that the Model Answer contains hallucinations with respect to the Prompt and the Expected Answer.
Definition of Hallucination for this task:
A hallucination occurs if the Model Answer contains:
1. Factual Inaccuracy: Statements that contradict the Expected Answer.
2. Internal Nonsense: The answer is self-contradictory, nonsensical, misleading, erroneous or otherwise semantically incoherent.
3. Unsupported Fabrication: Information (facts, details, conclusions) that is not contained in, cannot be inferred from, or contradicts the provided Prompt.
You must output a single floating-point number in the range [0.0, 1.0], representing the continuous likelihood of hallucination.
Important requirements:
-The score MUST be a continuous value, not a discrete category.
- Do NOT snap, round, or map your score to fixed anchor values.
- Avoid simple fractions such as 0.25, 0.5, 0.75 unless they are genuinely exact.
- Prefer nuanced values (e.g., 0.13, 0.42, 0.67) when appropriate.
- Use up to 5 decimal places.
Guidance for scoring (DO NOT copy numbers literally; use them as intuition):
- Answers that exactly match the Expected Answer should be 0.0.
- Answers that are mostly correct, with minor discrepancies, should be slightly higher.
- Answers that are partially supported, with some unclear or misleading facts, should be in the middle.
- Answers that are mostly unsupported or contain many misleading or incorrect facts should be high.
- Answers that are empty, only line breaks, completely nonsensical, misleading, erroneous or unrelated to the Prompt should be at the maximum.
Rules:
- Rephrasing or paraphrasing of the Expected Answer is NOT hallucination.
- Stylistic or formatting differences are NOT hallucination.
- Output MUST be valid JSON.
- Output MUST contain ONLY the JSON object, no explanations.
Analysis Steps (Consider in order):
1. Factual Accuracy: How closely does it match the Expected Answer's facts and sense?
2. Groundedness: Is the Model Answer supported by and relevant to the Prompt?
3. Fabrication: Does it introduce information absent from the Prompt?
4. Coherence: Is the answer self-consistent and logically structured?

*INPUT*
Prompt: {prompt}
Expected Answer: {expected_answer}
Model Answer: {answer}

---

Figure 1: Prompt template used to obtain hallucination scores from the judge LLM.

Let $\alpha, \beta \geq 0$ be weighting coefficients such that

$$\alpha + \beta = 1. \tag{5}$$

We define the aggregated target score as a normalized weighted sum:

$$\text{target\_score} = \alpha h + \beta(1 - s). \tag{6}$$

Here, $\alpha$ controls the contribution of the hallucination score, while $\beta$ determines the influence of semantic dissimilarity. The term $(1 - s)$ is used to align both components in the same direction: since $s$ measures cosine similarity, $(1 - s)$ corresponds to cosine distance Sahoo & Maiti (2025), ensuring that higher values consistently indicate a higher likelihood of hallucination. The constraint $\alpha + \beta = 1$ guarantees a normalized linear combination.

The final binary target variable is defined using a threshold $\tau$:

$$y = \begin{cases} 1, & \text{if target\_score} \geq \tau \\ 0, & \text{otherwise.} \end{cases} \tag{7}$$

In the experimental configuration, we set

$$\tau = 0.5. \tag{8}$$

Thus, responses with $\text{target\_score} \geq 0.5$ are assigned to the positive class (hallucinated), while the remaining samples are labeled as non-hallucinated. Adjusting $\alpha$ and $\beta$ allows controlling the relative sensitivity of the labeling procedure to hallucination severity versus semantic deviation from the reference answer.

After obtaining hallucination scores from Gemma and computing the cosine similarity values, we performed a grid search over $\alpha$ and $\beta$ in the interval $[0, 1]$ with step of 0.1. For each combination, we computed the positive class fraction. The corresponding heatmap is presented in Figure 2.

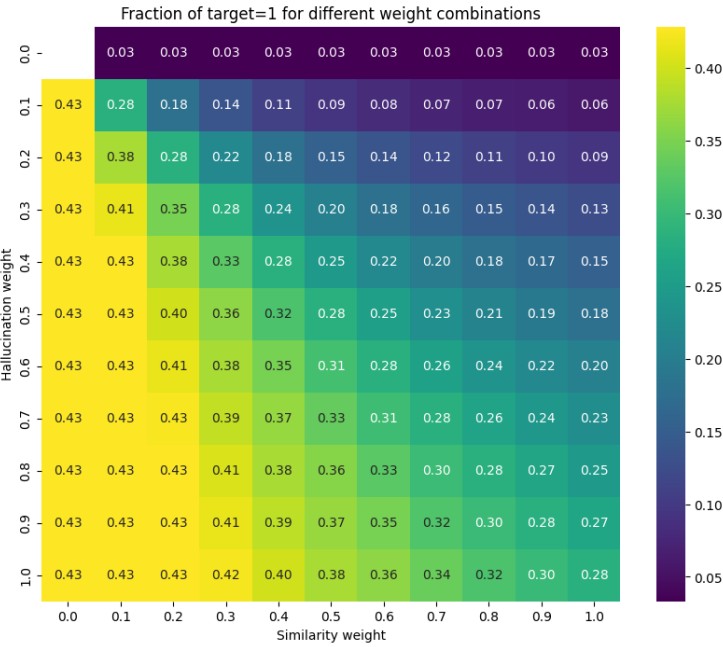

Figure 2: Positive class fraction for different weight combinations

As a result, we selected the coefficient values $\alpha = 0.6$ and $\beta = 0.4$, which yield approximately 35% of samples assigned to the positive class and 65% to the negative class. This distribution reflects a slight class imbalance avoiding extreme disproportionality preserving realistic asymmetry in the data. The chosen weighting is also supported by the intuition that the LLM-as-a-judge score should have a stronger influence than the semantic distance metric, while not fully determining the decision.

## 4.3 FINAL FEATURES EXTRACTION

As mentioned above, one of the core features expected to improve the performance of the classification pipeline are token log-probabilities of each generated response. However, classical machine learning algorithms require input in the form of fixed-length numerical feature vectors and cannot directly consume variable-length lists of floating-point numbers without first transforming them into a uniform representation suitable for model input Le & Mikolov (2014). It is necessary to derive a set of summary statistics that capture the essential characteristics of the log-probability distributions and can be represented as a set of features for downstream classification.

Let $\{\ell_i\}_{i=1}^n$ denote the token-level log-probabilities of a generated response.

As previously defined in Equations 1, 2, and 3, we compute the mean log-probability $\mu$, the standard deviation $\sigma$, and the fraction of low-confidence tokens $f$ as the separated features too. They characterize the overall confidence level, variance of token-level certainty, and the prevalence of highly uncertain tokens, respectively.

In addition to these features, we extract the following complementary ones.

Median log-probability

The median provides a robust estimate of central tendency that is less sensitive to extreme values than the mean:

$$\text{median}(\ell_1, \ldots, \ell_n) \tag{9}$$

Minimum log-probability:

$$\ell_{\min} = \min_{1 \le i \le n} \ell_i. \tag{10}$$

This feature captures the most uncertain token in the sequence and reflects the strongest local drop in model confidence during generation.

Maximum log-probability

$$\ell_{\max} = \max_{1 \le i \le n} \ell_i. \tag{11}$$

Although less directly related to hallucination risk, this statistic provides complementary information about peak confidence levels.

Normalized entropy of token probabilities

To measure the global variance of confidence across tokens, we convert log-probabilities into normalized weights:

$$w_i = \frac{e^{\ell_i}}{\sum_{j=1}^n e^{\ell_j}}. \tag{12}$$

Then we compute the Shannon entropy Shannon (1948):

$$H = -\sum_{i=1}^n w_i \log w_i. \tag{13}$$

To ensure comparability across sequences of different lengths, we normalize the entropy:

$$H_{\text{norm}} = \frac{H}{\log n}. \tag{14}$$

By construction,

$$0 \leq H_{\text{norm}} \leq 1. \tag{15}$$

Values close to $0$ indicate that confidence is concentrated in a small subset of tokens, whereas values close to $1$ correspond to a more uniformly distributed confidence pattern.

The correlation matrix between numeric features is shown in Figure 3.

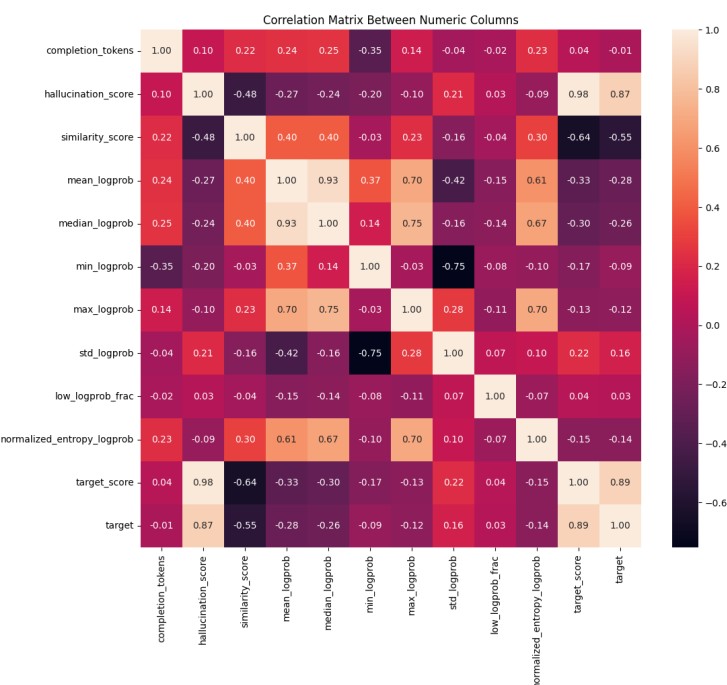

Figure 3: Correlation matrix between numberic features in the dataset

The correlation matrix does not reveal strong linear dependencies between the features, indicating the absence of pronounced multicollinearity. A moderate correlation can be observed between the target variable and the statistical descriptors of token log-probabilities. This suggests that log-probability-based features may carry informative signals relevant for classfication and are likely to contribute meaningfully to the performance of the trained models.

## 5 CLASSIFIERS TRAINING

This section outlines the process of selecting the best classification pipeline, which consists of both a textual feature vectorization component and a classification model. We evaluate different combinations of vectorization techniques and classifiers to determine the configuration that achieves the best performance on the target task and then tuning hyperparameters of best pipeline.

### 5.1 BEST CLASSIFICATION PIPELINE SELECTION

After successfully building the dataset we can finally train our classifier. As classification models not able to process raw textual data, we need to vectorize *prompt* and *answer* columns. As we work on building a lightweight model based on classical machine learning algorithm, we need to make a sparse embeddings rather than dense ones Joachims (1998). For this we applied two sparse vectorizing approaches – Bag-of-Words (Count Vectorizer) and TF-IDF and compared their performance with different classification algorithms.

After constructing the dataset, we proceed to the training of classification models. Since classical machine learning algorithms cannot directly operate on raw textual data, the *prompt* and *answer*

fields must first be transformed into numerical representations. As we work on building a lightweight classification pipeline based on traditional machine learning methods, we employ sparse vector representations rather than dense embeddings Joachims (1998). We applied two widely used sparse vectorization approaches: the Bag-of-Words model (Count Vectorizer) Harris (1954) and Term Frequency–Inverse Document Frequency (TF–IDF) weighting Salton & Buckley (1988). We evaluate both representations in combination with several classification algorithms to determine the most effective pipeline configuration.

As the classification models we compare the following algorithms: Dummy Classifier (predicts the objects as a most frequent class from training data), Logistic Regression Cox (1958) (as we solving the task of binary classification), Decision Tree classifier Breiman et al. (2017), Random Forest Breiman (2001) two gradient boosting implementations – LightGBM Ke et al. (2017) and CatBoost Prokhorenkova et al. (2018).

We evaluate all combinations of vectorizers and classifiers using an out-of-fold validation scheme Stone (1974) based on 5-fold cross-validation. To preserve the class distribution across folds, we employ stratified k-fold splitting Kohavi et al. (1995), which maintains approximately the same proportion of target classes in each fold.

Model performance is assessed using several standard metrics for binary classification. The primary evaluation metric is the Area Under the Receiver Operating Characteristic Curve (ROC AUC) Hanley & McNeil (1982), as it measures the model's ability to rank positive instances higher than negative ones independently of a fixed decision threshold. This is especially relevant for our task, where the classification threshold may be adjusted at inference time. Also we report F1-score van Rijsbergen (1979), precision, recall, accuracy, and the training time of each pipeline to compare computational efficiency. All experiments were conducted on a MacBook Pro M4 Max (36 GB RAM, 1 TB SSD).

We also applied stratified k-fold splitting in order to ensure that each fold gonna have the same target distribution. For classifiers quality evaluation we used the following metrics, applicable for task of binary classification: ROC AUC score (the most important metric for us as we want be able change positive class threshold during the inference), F1-score, precision, recall, accuracy and training time of each pipeline (all experiments were conducted on MacBook Pro M4 Max 36GB/1TB).

All classifications models were initialized with their default parameters as we aim to find most suitable one and then tune its hyperparameters. The performance metrics of the classification pipelines based on the Bag-of-Words (BoW) vectorizer are presented in Figure 4.

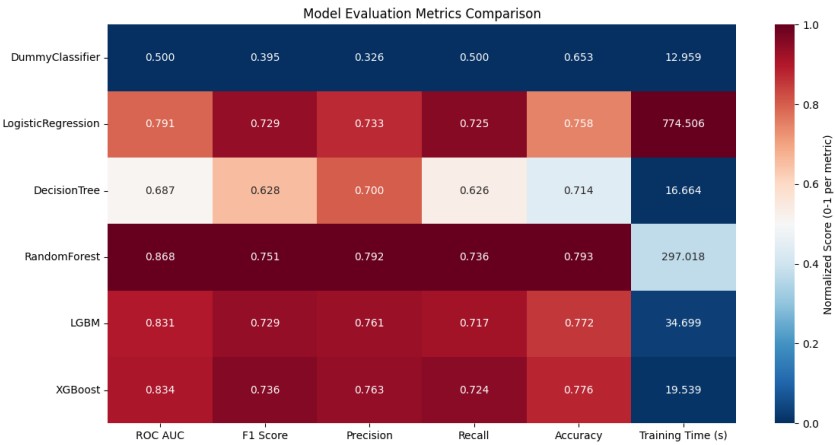

Figure 4: Metrics of pipelines based on BoW vectorization

The best performance was shown by tree-based classifiers: Random Forest and gradient boosting implementations. In Figure 5 the performance metrics of the classification pipelines based on the TF-IDF vectorizer are presented.

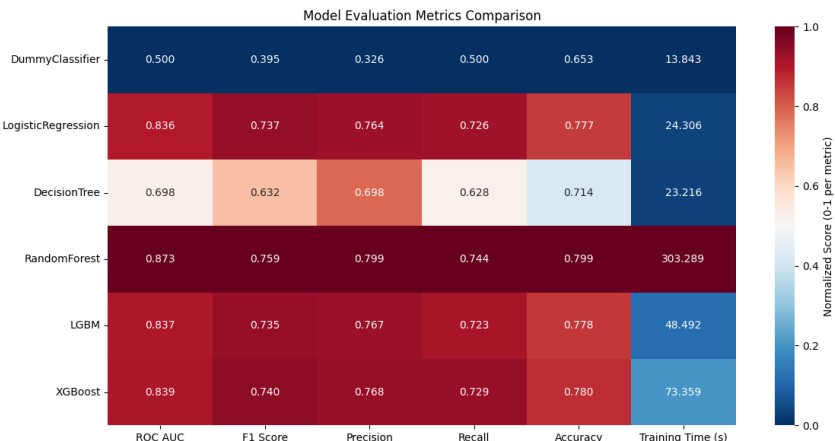

Figure 5: Metrics of pipelines based on TF-IDF vectorization

As we see, the best performance was shown by tree-based estimators too. A noticeable improvement can be observed for Logistic Regression. This indicates that the TF-IDF representation provides a more informative and discriminative vectorization of textual features compared to the alternative sparse encoding.

## 5.2 BEST CLASSIFIER TUNING

On the previous step we found, that the best performative pipeline for current task is the combination of TF-IDF vectorizer and Random Forest. In this section we describe its tuning process.

In the previous stage, we identified the combination of TF-IDF vectorization and Random Forest as the most effective pipeline for the current task. In this section, we describe the hyperparameter tuning procedure for this model.

We tune the following hyperparameters:

*Number of trees* ($n_{\text{estimators}}$). This parameter controls the size of the ensemble. We consider the following values:

$$n_{\text{estimators}} \in \{20, 50, 100, 300\}.$$

Increasing the number of trees generally reduces variance and improves stability, at the cost of higher computational time.

*Maximum tree depth* (max_depth). This parameter limits the depth of each decision tree:

$$\text{max\_depth} \in \{\text{None}, 5, 10, 20\}.$$

Restricting tree depth reduces model complexity and helps prevent overfitting, while allowing unlimited depth (None) enables fully grown trees.

*The splitting criterion* that determines how node impurity is measured during tree construction. We evaluate three commonly used criteria:

- Gini impurity,
- entropy (information gain),
- log-loss.

We used Halving Search as the best parameters search algorithm and ROC AUC as the optimizing metric because this is the target one for our task.

For hyperparameter optimization, we employ the Successive Halving search strategy Li et al. (2018). As the optimization objective, we use ROC AUC, since it serves as the primary evaluation metric for our task.

Successive Halving is a resource-efficient hyperparameter search algorithm that iteratively allocates increasing computational resources to the most promising candidate configurations while eliminating poorly performing ones at early stages. This process continues until a small set of high-performing configurations remains. Compared to exhaustive grid search, Successive Halving significantly reduces computational cost while maintaining competitive selection quality.

After completing the search the following hyperparameters were selected:

- $criterion$ – Gini,
- $max\_depth$ – None,
- $n_{estimators}$ – 300.

The confusion matrix and ROC curve of the best pipeline is shown in Figure 6.

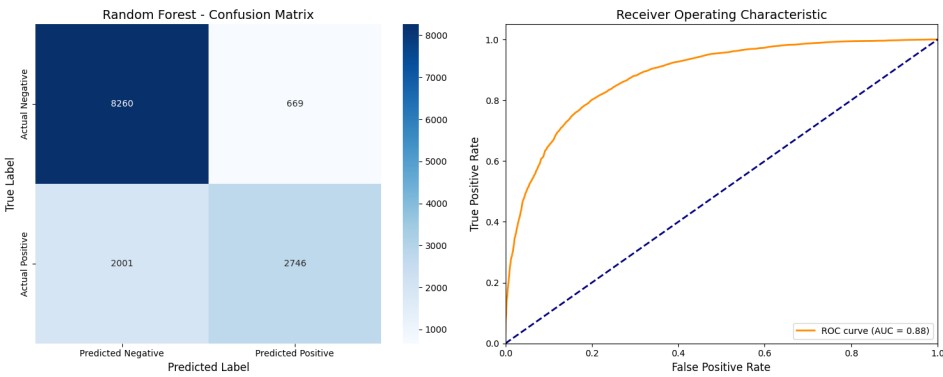

Figure 6: Confusion matrix and ROC curve of the best classification pipeline

Parameters tuning increased the ROC AUC of the pipeline to **0.88** which is a decent metric for such lightweight classifier.

## 6 FUTURE WORK

After successfully developing a lightweight hallucination detector, we plan to integrate it as an output guardrail within the multi-agent architecture of the Donetsk State University intelligent assistant.

Two primary integration strategies are considered. The first approach is to deploy the trained classifier directly as a post-generation filtering module, flagging potentially unreliable responses before they are delivered to the end user. The second one involves extending the training dataset by incorporating generations produced by LLaMA 3.1 8B Instruct[1] and re-running the full training and evaluation pipeline described in this study.

## 7 CONCLUSION

In this paper, we presented a process of labeling LLM-generated responses with respect to hallucination presence, given a prompt and an expected answer. Based on this labeled dataset, we evaluated multiple lightweight classification pipelines combining sparse textual vectorization methods (Bag-of-Words and TF-IDF) with classical machine learning algorithms. The novelty of our study lies in using not only textual features for determining hallucinations in the responses, but also by incorporating token log-probabilities of the response (presented by extracted scalar features from them).

The best-performing pipeline – TF-IDF combined with a tuned Random Forest classifier achieved a ROC AUC score of **0.88**, demonstrating that lightweight models can provide strong performance in hallucination detection without relying on large neural architectures. The feature importances of the best classification pipeline are shown in Figure 7.

---

[1]https://huggingface.co/meta-llama/Llama-3.1-8B-Instruct

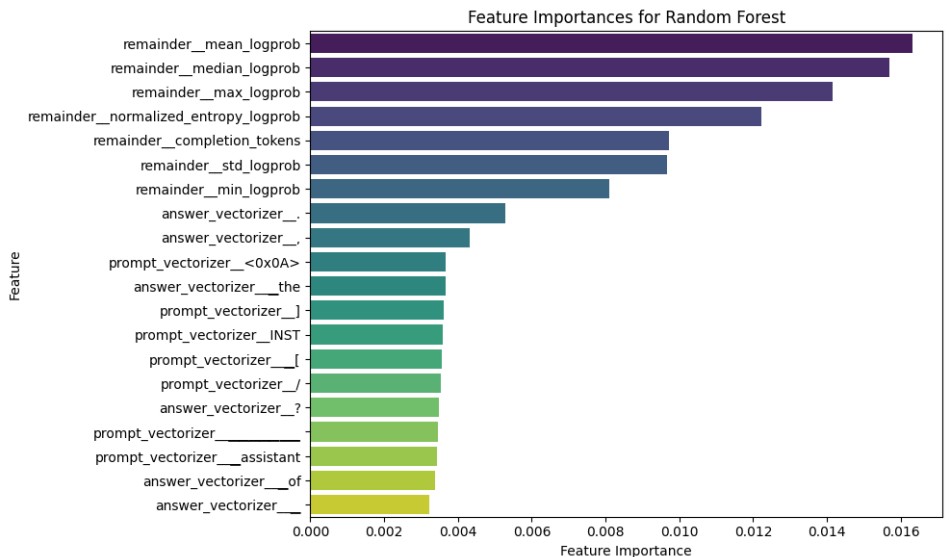

Figure 7: Top-20 features with their importances of the best classification pipeline

Analysis of feature importances reveals that the features derived from token log-probabilities significantly outperform pure textual ones, confirming our hypothesis that generation-time uncertainty signals play a crucial role in hallucination detection. These findings highlight the effectiveness of combining sparse textual representations with probabilistic confidence-based features in building efficient and interpretable hallucination detection systems.

ACKNOWLEDGMENTS

The study was carried out with the financial support of the Ministry of Education and Science of the Russian Federation within the framework of the state task on the topic "Development and improvement of intelligent classification and forecasting methods for pattern recognition and modeling of information processes" FREM-2024-0001 (Registration number 1023111000141-9-1.2.1)

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
