# OpenReview forum: "Detecting Hallucinations in LLM Responses Using Token-Level Log-Probability Signals"
_mathai.club/MathAI/2026/Conference — 2026 Oral_

### Official Review · Reviewer_w66m · 2026-03-10
**Lightweight hallucination detection using token-level uncertainty signals: promising idea but methodological concerns**

**Rating:** 6
**Confidence:** 3

**Review:**

This paper proposes a lightweight approach to detecting hallucinations in large language model (LLM) outputs by combining token-level log-probability signals with textual features. The authors construct a dataset from OpenOrca prompts and Mistral-7B-generated responses and introduce statistical features derived from token log-probabilities, including mean, variance, entropy, minimum/maximum values, and the fraction of low-confidence tokens.

A semi-automatic labeling pipeline is used to assign hallucination labels by combining (1) an LLM-as-a-judge hallucination score produced by Gemma-3 and (2) cosine similarity between the generated answer and the reference answer. Classical machine learning classifiers are then trained on sparse textual features (BoW and TF-IDF) combined with the log-probability statistics. The best-performing pipeline (TF-IDF + Random Forest) reportedly achieves a ROC-AUC of approximately 0.88.

The work is motivated by the need for efficient hallucination detection components in latency-sensitive multi-agent systems.

## Strengths



**1. Practical motivation**

The paper addresses an important practical problem: detecting hallucinations in LLM outputs. The focus on lightweight detectors suitable for real-time guardrail systems is well motivated, particularly for deployment in multi-agent or RAG-based architectures.

**2. Use of generation-time uncertainty signals**

The use of token-level log-probability statistics as features is a reasonable and promising direction. Such signals capture model uncertainty during generation and may contain useful information about hallucination risk.

**3. Clear and structured pipeline**

The methodology is organized clearly and includes:

- dataset preprocessing and reduction
- semi-automatic labeling
- feature extraction from token log-probabilities
- evaluation of multiple classifiers
- hyperparameter tuning

The overall workflow is easy to follow and appears reproducible.

**4. Efficiency considerations**

The decision to rely on classical machine learning models rather than heavy neural architectures is appropriate for real-time systems where inference latency is critical.

**5. Comparative experiments**

The paper compares several classification algorithms (Logistic Regression, Decision Tree, Random Forest, LightGBM, CatBoost) and vectorization approaches (BoW and TF-IDF) using cross-validation and multiple evaluation metrics.

## Weaknesses

Despite the interesting idea, there are several methodological issues that weaken the empirical claims.

**1. Potential selection bias in dataset construction**

The dataset reduction procedure relies on a *risk score* computed from token log-probability statistics (mean log probability, variance, and fraction of low-confidence tokens). This score is used to select examples with the highest, lowest, and median risk per prompt.

However, the final classification model uses **features derived from the same log-probability statistics**.

This creates a potential form of **selection bias or feature leakage**, since the dataset was constructed using signals closely related to those used by the model during training. As a result, the classifier may partially learn patterns introduced by the filtering procedure rather than genuine hallucination characteristics. This could inflate the reported ROC-AUC.

A more robust evaluation would avoid filtering examples using signals that later appear as model features.

**2. Labeling methodology may introduce noise**

The hallucination labels are generated automatically using a combination of:

- an LLM-as-a-judge score (Gemma-3)
- cosine similarity between generated and reference answers.

While this approach is practical, it raises several concerns:

- LLM-as-a-judge systems are known to produce noisy and biased evaluations.
- Cosine similarity does not reliably capture factual correctness.
- No human validation of the labeling procedure is reported.

Without at least partial manual validation, the reliability of the ground truth labels remains uncertain.

**3. Limited evaluation scope**

The experiments rely on responses generated by **only one base model (Mistral-7B)** and on a single dataset derived from OpenOrca.

Hallucination detectors are known to be highly model-dependent. A stronger evaluation would include:

- cross-model evaluation (e.g., training on Mistral and testing on LLaMA or GPT models)
- multiple datasets or domains
- robustness checks under different generation parameters.

Without these tests it is unclear whether the detector generalizes beyond the specific dataset used in this work.


**4. Lack of comparison with recent hallucination detection approaches**

The paper evaluates several classical classifiers but does not compare against stronger baselines from recent literature such as neural hallucination detectors or uncertainty-based detection methods.

Including such comparisons would help position the contribution relative to the state of the art.

**5. Writing quality**

The paper contains a noticeable number of grammatical errors, repeated phrases, and stylistic inconsistencies. While the overall structure is understandable, the writing quality should be improved for clarity.



## Suggestions for Improvement

1. Validate the labeling pipeline using human annotations on a subset of the dataset.
2. Avoid using log-probability–derived signals for both dataset filtering and feature extraction, or explicitly evaluate the impact of this design choice.
3. Evaluate cross-model generalization to demonstrate robustness.
4. Compare the method with stronger hallucination detection baselines from recent work.
5. Include ablation studies isolating the contribution of textual features vs. log-probability features.
6. Improve proofreading and language quality.

---

## Overall Assessment

The paper proposes a practical and potentially useful idea: combining token-level uncertainty signals with lightweight machine learning models for hallucination detection. The approach is computationally efficient and could be valuable in real-time guardrail systems.

However, concerns about the dataset construction procedure, labeling reliability, and limited evaluation reduce confidence in the reported results. Addressing these issues would significantly strengthen the work.

---

### Official Review · Reviewer_bAZa · 2026-03-11
**Review of "Detecting Hallucinations in LLM Responses Using Token-Level Log-Probability Signals"**

**Rating:** 7
**Confidence:** 2

**Review:**

This paper addresses the practical problem of detecting hallucinations in LLM-generated responses, with a focus on building a lightweight classifier suitable as an output guardrail in latency-sensitive multi-agent systems. The authors construct a labeled dataset by combining LLM-as-a-judge scores (Gemma-3 12B) with cosine similarity between generated and reference answers. They extract features from both the text (using sparse vectorizers like BoW and TF-IDF) and token-level log-probabilities (aggregated into statistics such as mean, standard deviation, entropy). After comparing several classical ML algorithms, they select a TF‑IDF + Random Forest pipeline and optimize its hyperparameters via successive halving, achieving a ROC AUC of 0.88. The work is open-sourced and clearly documented.

---

**Quality and Clarity (8/10)**

The paper is well-structured and clearly written. The motivation, methodology, and experiments are presented in a logical order, making it easy to follow. Key steps are described with sufficient detail to ensure reproducibility: the risk-based dataset filtering (Equation 4), the semi-automatic labeling procedure with $\alpha$ and $\beta$ weights, the extraction of log-probability features (Equations 1–3, 9–14), and the cross-validation setup. Figures (correlation matrix, performance comparisons, confusion matrix) and tables effectively support the text. Minor issues include a slightly redundant paragraph on metrics (page 11) and a lack of justification for the manually chosen coefficients in the risk function ($\alpha=1,\beta=0.5,\gamma=0.3,\tau=-5.0$). Overall, the exposition is professional and accessible.

**Originality (6/10)**

The work does not introduce a novel algorithmic concept; rather, its originality lies in the **system-level combination** of existing ideas to solve a specific practical problem. The use of log-probability statistics as features for hallucination detection has been explored before, and sparse text representations (TF‑IDF) with tree-based classifiers are standard tools. However, the paper’s contribution is in demonstrating that this lightweight combination can achieve competitive performance (ROC AUC 0.88) without resorting to neural networks or additional LLM calls. This pragmatic, engineering-focused approach is valuable for deployment scenarios where latency and compute resources are constrained. The systematic comparison of multiple classifiers and vectorizers, along with a thoughtful data reduction strategy (risk score), adds to the practical contribution.

**Significance (7/10)**

The significance of this work is primarily practical. It provides a ready-to-use, fast hallucination detector that can be integrated as a guardrail in multi-agent systems. The achieved ROC AUC of 0.88 on the test set suggests that the model can effectively rank hallucinated responses higher than truthful ones, allowing a tunable threshold at inference time. The open-source release of the code further enhances its potential impact. However, the generalizability of the model remains uncertain: it was trained on responses from Mistral 7B Instruct on the Open Orca dataset. Whether the same features and model would perform equally well on other LLMs (e.g., LLaMA-3, GPT-4) or on different domains is not evaluated. The authors acknowledge this in the Future Work section, but it limits the immediate broader significance.

---

**Detailed Comments**

1.  **Data Labeling**: The semi-automatic labeling using a combination of an LLM judge ($h$) and semantic similarity ($s$) is a sensible approach given the lack of ground truth. The grid search over $\alpha$ and $\beta$ to achieve a 35%/65% class split is a good practice. However, the final threshold $\tau=0.5$ for binarization is somewhat arbitrary; a sensitivity analysis of this threshold on the downstream classifier’s performance would have strengthened the work.

2.  **Feature Engineering**: The set of features derived from log-probabilities is comprehensive (mean, std, fraction low-confidence, median, quantiles, entropy, normality index). The correlation matrix (Figure 3) shows only moderate correlations with the target, indicating that these features indeed add non-redundant information. The feature importance analysis (Figure 7) confirms that log-probability statistics (especially mean and std) contribute meaningfully alongside textual tokens, validating the authors’ hypothesis.

3.  **Experimental Setup**: The comparison of multiple classifiers with default hyperparameters is appropriate for selecting a baseline. Using out-of-fold validation with stratified 5-fold CV ensures robust estimates. The subsequent hyperparameter tuning with successive halving is efficient. Reporting training time on a MacBook Pro provides a useful reference for practitioners concerned about latency.

4.  **Missing Comparisons**: The paper would be strengthened by comparing the proposed pipeline against other lightweight hallucination detectors (e.g., HaluNet, or even a simple threshold-based detector using only log-probability features). This would contextualize the 0.88 ROC AUC within the landscape of existing efficient methods.

5.  **Writing**: The text is generally polished, but a few minor errors exist (e.g., duplicate paragraph on metrics, page 11). The references are relevant and up-to-date.

---

**Strengths**

*   **Practical focus**: The work directly addresses a real-world need for low-latency hallucination detection in multi-agent systems.
*   **Reproducible pipeline**: Detailed description of data preprocessing, feature extraction, and model selection, accompanied by open-source code.
*   **Solid empirical results**: The final model achieves a strong ROC AUC of 0.88 with a lightweight Random Forest + TF‑IDF combination.
*   **Clear demonstration of feature value**: Log-probability statistics are shown to be informative beyond pure text features.

**Weaknesses**

*   **Limited novelty**: The approach is a combination of known techniques rather than a new algorithmic advance.
*   **Potential labeling bias**: The ground truth depends on an LLM judge (Gemma-3), which may introduce systematic biases.
*   **Uncertain generalizability**: The model is trained and evaluated on data from a single LLM (Mistral 7B) and a single source dataset (Open Orca). Its performance on other models/domains is unknown.
*   **Lack of comparison with strong lightweight baselines**: It is unclear how the proposed method stacks up against other efficient detectors.

---

### Official Review · Reviewer_d7ce · 2026-03-13
**Useful practical contribution with promising results, though the validation could be stronger**

**Rating:** 7
**Confidence:** 3

**Review:**

This paper tackles a practically important problem: detecting hallucinated LLM outputs with a lightweight model that could be used as a guardrail in latency-sensitive systems. The basic direction is reasonable. Using token-level log-probability summaries together with sparse text features is an intuitively plausible engineering approach, and the emphasis on deployability is a real strength.

However, I think the empirical validation is somewhat weaker than the practical contribution of the paper. My main concern is that the reported `ROC AUC = 0.88` is not yet as easy to interpret as one would like, because the dataset construction, pseudo-labeling procedure, and evaluation protocol introduce some potential sources of bias. Even so, I view this as a useful engineering contribution with clear practical relevance, and the weaknesses seem more like limitations to address in a revision than reasons to reject the work.

**Main concerns**

1. **The evaluation protocol could be stronger, especially around prompt-level splitting.**
   The source dataset contains multiple generations per prompt, and even after reduction the paper keeps several samples from the same prompt: the deterministic sample, the lowest-risk sample, the highest-risk sample, and the median-risk sample. The evaluation is then described as ordinary stratified 5-fold cross-validation over rows. This raises a concern that correlated examples from the same prompt may appear in both train and validation folds. A grouped split by prompt would make the reported AUC more convincing and would better establish generalization.

2. **The target labels are heuristic and would benefit from stronger validation.**
   A response is labeled using a weighted combination of an LLM-as-a-judge hallucination score and cosine dissimilarity to a single reference answer. The paper then selects the weights because they yield roughly a `35% / 65%` class split. This is a practical but somewhat heuristic target-definition procedure. Without human annotation or at least a calibration study, it remains unclear how closely the learned detector aligns with a stronger notion of hallucination.

3. **The data reduction step may make the contribution of token-level features look cleaner than it really is.**
   Before training, the authors downsample the full dataset using a hand-crafted risk score based on the same family of signals that later form the core features: mean log-probability, standard deviation, and the fraction of low-confidence tokens. Even if the exact scalar risk score is removed later, the training/evaluation distribution has already been conditioned on the same uncertainty signals that the paper later emphasizes. This does not invalidate the approach, but it does make the final evidence less clean than it could be.

4. **The label definition may penalize valid alternative answers.**
   One component of the target is cosine similarity to a single reference answer. For many prompts, especially in open-ended instruction-following data, multiple materially correct answers are possible. A response can be factually adequate while still differing semantically from the dataset's preferred answer. This means the current setup may sometimes conflate hallucination with deviation from one reference formulation.

5. **The ablation story could be clearer and more complete.**
   The abstract says the study performs ablations quantifying the contribution of token-level signals versus text-only features. In the body, I do not see a fully clean ablation that isolates `text-only`, `logprob-only`, and `combined` variants under the same evaluation protocol. Model comparisons and feature-importance plots are informative, but a more explicit ablation would make the contribution easier to assess.

6. **The final evaluation protocol could be presented more rigorously.**
   The paper describes stratified cross-validation for model comparison and then successive-halving hyperparameter tuning, but it does not clearly present a truly held-out test set or a nested evaluation protocol. A more explicit separation between model selection and final evaluation would strengthen confidence in the reported `0.88` ROC AUC.

7. **The deployment framing is somewhat broader than the current evidence.**
   The paper repeatedly positions the method as a guardrail for multi-agent systems and high-stakes domains, but the experiments are limited to responses from a single generator model on a single source dataset, with a fixed `top-p` and only two temperature values. There is no cross-model evaluation, no cross-domain transfer, no robustness study across decoding settings, and no downstream analysis showing that the detector actually improves system reliability. The paper also emphasizes latency, but reports training time rather than inference-time latency or end-to-end system cost.

8. **The term "early detection" feels somewhat overstated.**
   The proposed features are computed over the full generated response. That supports post-generation filtering more directly than true online interruption during generation. A slightly narrower framing would make the contribution more precise.

9. **The connection to MathAI could be made more explicit.**
   As submitted, this reads primarily as a general NLP/LLM safety engineering paper. The paper would benefit from a clearer explanation of why this contribution belongs specifically in the MathAI venue.

10. **The manuscript quality still needs polishing.**
   There are grammatical errors, duplicated sentences, and some editorial sloppiness. These are not central flaws, but they do make the paper feel less mature than the underlying idea deserves.

**Strengths**
- The paper addresses a real deployment problem that matters in practice.
- The lightweight, classical-ML framing is sensible for latency-sensitive settings.
- Token-level uncertainty signals are worth studying, and the overall system idea is easy to understand.

**Weaknesses**
- The evaluation protocol could be stronger, especially around prompt-grouped splitting.
- The target labels are heuristic and not validated against human annotation.
- The dataset is preselected using the same signal family that later drives the model.
- The ablation and generalization evidence is weaker than the deployment framing suggests.

**Suggestions for improvement**
- Redo the evaluation with **grouped splits by prompt** at minimum; ideally include a truly held-out prompt set.
- Validate the pseudo-labeling pipeline against human annotations on a reasonably sized subset.
- Report clean ablations for `text-only`, `logprob-only`, and `combined` features.
- Compare against a very simple baseline such as thresholding the handcrafted risk score itself.
- Test cross-model transfer and robustness across decoding settings.
- Narrow the framing from broad multi-agent guardrail claims to what is actually demonstrated.

**Overall assessment**
The paper has a practical intuition and, in my view, a useful applied contribution. I also think the lightweight deployment angle is genuinely relevant, and the core empirical result is promising enough to merit acceptance. My reservations are mainly about how broadly the current validation supports the paper's claims, not about whether there is a worthwhile contribution here. With stronger evaluation protocol choices and clearer ablations, this could become a much stronger paper, but even in its current form I think the practical value outweighs the methodological weaknesses.

---

### Decision · Program_Chairs · 2026-03-14

**Decision:**

Accept (Oral)

**Comment:**

Dear Author(s),

On behalf of the Program Committee of the International Conference on Mathematics of Artificial Intelligence (MathAI 2026), we are pleased to inform you that your paper has been accepted for an oral presentation at MathAI 2026.

Your paper was evaluated through a rigorous two-stage review process involving both automated screening and expert review by members of the Program Committee. The reviewers recognized the quality and contribution of your work.

Presentation details:

- Format: Oral presentation (15–20 minutes + 5 minutes Q&A)
- Mode: You may present either in person (offline) at the conference venue in Sirius, Russia, or remotely via Zoom. Please indicate your preferred mode when confirming your participation.
- Conference dates: Marh 30 - April 3, 2026
- Website: https://mathai.club

Next steps:

1. Please confirm your participation and presentation mode by replying to this email mathai.club@yandex.ru no later than March 15, 2026 18:00 Moscow time.
2. If you plan to attend in person, the organizing committee will provide accommodation details separately.
3. Please prepare your final camera-ready manuscript according to the formatting guidelines available at https://mathai.club and upload it to OpenReview by March 15, 2026 18:00 Moscow time.

Should you have any questions regarding the program, logistics, or your presentation slot, please do not hesitate to contact us.

We look forward to your contribution to MathAI 2026.

With kind regards,

MathAI 2026 Program Committee
International Conference on Mathematics of Artificial Intelligence
https://mathai.club
OpenReview: https://openreview.net/group?id=mathai.club/MathAI/2026/Conference
Telegram: https://t.me/MathAI_club
Email: mathai.club@yandex.ru